

# Propofol inhibits stromatoxin-1-sensitive voltage-dependent K$^+$ channels in pancreatic β-cells and enhances insulin secretion

Munenori Kusunoki[1,2], Mikio Hayashi[3], Tomohiro Shoji[1,2], Takeo Uba[1,2], Hiromasa Tanaka[2], Chisato Sumi[1,2], Yoshiyuki Matsuo[2] and Kiichi Hirota[2]

[1] Department of Anesthesiology, Kansai Medical University, Hirakata, Japan
[2] Department of Human Stress Response Science, Institute of Biomedical Science, Kansai Medical University, Hirakata, Japan
[3] Department of Cell Physiology, Institute of Biomedical Science, Kansai Medical University, Hirakata, Japan

Corresponding author
Kiichi Hirota, hif1@mac.com

## ABSTRACT

**Background**. Proper glycemic control is an important goal of critical care medicine, including perioperative patient care that can influence patients' prognosis. Insulin secretion from pancreatic β-cells is generally assumed to play a critical role in glycemic control in response to an elevated blood glucose concentration. Many animal and human studies have demonstrated that perioperative drugs, including volatile anesthetics, have an impact on glucose-stimulated insulin secretion (GSIS). However, the effects of the intravenous anesthetic propofol on glucose metabolism and insulin sensitivity are largely unknown at present.

**Methods**. The effect of propofol on insulin secretion under low glucose or high glucose was examined in mouse MIN6 cells, rat INS-1 cells, and mouse pancreatic β-cells/islets. Cellular oxygen or energy metabolism was measured by Extracellular Flux Analyzer. Expression of glucose transporter 2 (GLUT2), potassium channels, and insulin mRNA was assessed by $q$RT-PCR. Protein expression of voltage-dependent potassium channels (Kv2) was also assessed by immunoblot. Propofol's effects on potassium channels including stromatoxin-1-sensitive Kv channels and cellular oxygen and energy metabolisms were also examined.

**Results**. We showed that propofol, at clinically relevant doses, facilitates insulin secretion under low glucose conditions and GSIS in MIN6, INS-1 cells, and pancreatic β-cells/islets. Propofol did not affect intracellular ATP or ADP concentrations and cellular oxygen or energy metabolism. The mRNA expression of GLUT2 and channels including the voltage-dependent calcium channels Cav1.2, Kir6.2, and SUR1 subunit of $K_{ATP}$, and Kv2 were not affected by glucose or propofol. Finally, we demonstrated that propofol specifically blocks Kv currents in β-cells, resulting in insulin secretion in the presence of glucose.

**Conclusions**. Our data support the hypothesis that glucose induces membrane depolarization at the distal site, leading to $K_{ATP}$ channel closure, and that the closure of Kv channels by propofol depolarization in β-cells enhances Ca$^{2+}$ entry, leading to insulin secretion. Because its activity is dependent on GSIS, propofol and its derivatives are potential compounds that enhance and initiate β-cell electrical activity.

## INTRODUCTION

Proper glycemic control is one of the most important goals of patient management in critical care medicine including perioperative care (*Lipshutz & Gropper, 2009*; *Martinez, Williams & Pronovost, 2007*). A number of studies have demonstrated that hyperglycemia is one of the most serious risk factors for morbidity and mortality in critical care medicine (*Lipshutz & Gropper, 2009*). The intricate balance between the production and consumption of glucose is affected by blood glucose concentration, which determines the secretion of and sensitivity to insulin. However, external factors, including stress by surgical procedures and the drugs administrated for anesthetic management, can largely affect this internal balance (*Martinez, Williams & Pronovost, 2007*).

A line of animal and human studies have demonstrated that perioperative drugs, including volatile anesthetics such as halothane, enflurane, isoflurane, and sevoflurane, have an impact on glucose-stimulated insulin secretion (GSIS) (*Kitamura et al., 2009*; *Suzuki et al., 2015*). *Suzuki et al. (2015)* demonstrated a molecular mechanism behind the impairment of insulin secretion by isoflurane and sevoflurane.

One study showed the intravenous anesthetic pentobarbital induces whole-body insulin resistance in rats (*Tanaka et al., 2009*). Propofol (2,6-diisopropylphenol) is an intravenous anesthetic used for the induction and maintenance of anesthesia during surgical procedures and for long-term sedation of patients in intensive care units (*Sebel & Lowdon, 1989*; *Vasileiou et al., 2009*). Propofol has also been shown to cause insulin resistance in rats (*Sebel & Lowdon, 1989*). However, the specific mechanism underlying this phenomenon remains unknown (*Kim et al., 2014*; *Yasuda et al., 2013*). A number of studies have shown that propofol affects glucose metabolism and insulin sensitivity (*Kim et al., 2014*; *Sato et al., 2013*; *Tanaka et al., 2011a*). On the other hand, other studies indicate that its impact on glucose and insulin metabolism is not prominent under clinical settings in which various parameters are involved (*Lou et al., 2015*; *Maeda et al., 2018*; *Tanaka et al., 2011a*).

Insulin secretion from pancreatic β-cells in response to elevated blood glucose concentration plays a critical role in glycemic control (*Ashcroft, 2005*). It is reported that propofol anesthesia enhances insulin secretion and concomitantly exaggerates insulin resistance, compared with sevoflurane anesthesia (*Li et al., 2014*). Another report indicates that glucose levels in rats anesthetized with sevoflurane were significantly higher than those in rats anesthetized with propofol, and insulin levels in rats anesthetized with sevoflurane were significantly lower than those in rats anesthetized with propofol (*Kitamura et al., 2012*). However, molecular mechanisms behind the phenomenon are largely unknown at this moment. In this study, using cell biological and electrophysiological methods, we investigated the impact of propofol on insulin secretion at low and high glucose levels. In a series of experiments, we examined the effects of propofol on basal and glucose-stimulated insulin secretion and demonstrated that propofol, at clinically relevant doses, inhibits

stromatoxin-1-sensitive potassium channels (Kv) and facilitates insulin secretion (IS) in MIN6, and INS-1 cells, and pancreatic β-cells/islets.

## MATERIALS AND METHODS

### Cells and cell culture

Mouse insulinoma MIN6 cell lines were cultured in Dulbecco's modified Eagle's medium (DMEM) (Gibco, Grand Island, NY, USA) containing 450 mg/dl glucose, 10% fetal bovine serum (FBS), 50 μM β-mercaptoethanol, 100 U/ml penicillin, and 0.1 mg/ml streptomycin (Miyazaki, 1990 #103). Rat INS-1 cells were cultured in RPMI1640 (Sigma-Aldrich, St Louis, MO, USA), supplemented with 10% FBS, 10 mM HEPES, 2 mM L-glutamine, 1 mM sodium pyruvate, 50 μM β-mercaptoethanol, 100 U/ml penicillin, and 0.1 mg/ml streptomycin (*Asfari et al., 1992*).

### Isolation of mouse pancreatic islets

Male C57BL/6JJcl mice (8–10 weeks old, $n = 8$) were sacrificed by cervical dislocation in accordance with protocols approved by the Animal Experimentation Committee, Kansai Medical University (#19-088). Pancreatic islets were isolated by enzymatic digestion from the pancreas with a slight modification (*Lacy & Kostianovsky, 1967*). The pancreas was removed and digested with collagenase (Type IV, 195 U/ml; Worthington Biochemical, Lakewood, NJ, USA) in a solution containing 2 mM glucose and trypsin inhibitor (0.01%; Sigma-Aldrich) at 37 °C for 30 min with vigorous shaking. The pancreatic tissue was triturated with a pipette and washed two times with enzyme-free solution. Islets were selected with a glass micropipette under a stereomicroscope. Batches of ten islets were used for measurement of insulin concentration.

### Reagents

Propofol (2,6-diisopropylphenol) and 2,4-diisopropylphenol were obtained from Sigma Aldrich. Glibenclamide was obtained from Wako Pure Chemical Industries, Ltd. (Osaka, Japan), diazoxide from Abcam (Cambridge, MA, USA), and stromatoxin-1 from Alomone Labs Ltd. (Jerusalem, Israel).

### Measurement of insulin concentration

Insulin concentration in the culture medium of MIN6, INS-1 cells, and pancreatic β-cells/islets was assessed using the Mouse/Rat Insulin H-type enzyme-linked immunosorbent assay kit (Shibayagi Co. Ltd., Shibukawa, Japan), following the manufacturer's instructions (*Suzuki et al., 2015*). Detailed protocols are available in the Supplemental Information and at protocols.io (10.17504/protocols.io.v63e9gn).

### Cell growth assay

Cell growth was assessed using the CellTiter 96 AQueous One Solution Cell Proliferation Assay (Promega, Madison, WI, USA) (*Sumi et al., 2018b*; *Suzuki et al., 2015*). Cells were seeded at a density of $3 \times 10^3$ cells/well in 96-well plates and incubated for 0, 4, and 12 h. Cell viability was determined by comparing the absorbance values of the treated cells with that of the control cells (MIN6 cells at 24 h incubation), with the latter defined as 100%.

All experiments were done in triplicate or quadruplicate. Detailed protocols are available as Supplemental Information and at protocols.io (10.17504/protocols.io.v64e9gw).

## Caspase activity assays

The activities of caspase-3 and caspase-7 were determined using an Apo-ONE Homogeneous Caspase-3/7 Assay Kit (Promega), according to the manufacturer's protocols (*Okamoto et al., 2016*; *Sumi et al., 2018b*). In brief, cells were seeded at $2 \times 10^4$ cells/well on 96-well plates and incubated overnight. Cells were then treated with the experimental concentrations of camptothecin and propofol for varying lengths of time. Caspase activities were determined by comparing the luminescence values of the treated cells with those of the control cells (incubated without drugs), with the latter defined as 100%. All assays were conducted in triplicate and repeated at least twice. Detailed protocols are available as Supplemental Information and at protocols.io (10.17504/protocols.io.v7je9kn).

## ATP assay

Intracellular ATP content was evaluated using the CellTiter-Glo luminescent cell viability assay kit (Promega) (*Sumi et al., 2018a*). In brief, cells were seeded at a density of $3 \times 10^3$ cells/well on 96-well plates and allowed to grow for 1, 4, and 8 h in the either presence or absence of propofol. The relative ATP levels were determined by comparing the luminescence values of the treated cells to those of control cells cultured in 2 mM glucose. Assays were done in triplicate and repeated at least twice. Detailed protocols are available in the Supplemental Information and at protocols.io (10.17504/protocols.io.v7ke9kw).

## ADP assay

Intracellular ADP content was evaluated using the ADP Colorimetric/Fluorometric Assay kit (Abcam, Cambridge, MA, USA). In brief, cells were seeded at a density of $6 \times 10^5$ cells/well on 12-well plates and allowed to grow for 1, 4 and 8 h in the either presence or absence of propofol. The relative ADP levels were determined by comparing the luminescence values of the treated cells to those of control cells. Assays were done in triplicate and repeated at least twice. Detailed protocols are available in the Supplemental Information and at protocols.io (10.17504/protocols.io.zk4f4yw).

## Measurement of cellular oxygen consumption and extracellular acidification

The cellular oxygen consumption rate (OCR) and extracellular acidification rate (ECAR) were measured using an XFp Extracellular Flux Analyzer (Agilent Technologies, Santa Clara, CA, USA) (*Sumi et al., 2018a*). MIN6 cells were seeded at a density of $1 \times 10^4$ cells/well on the XFp Cell Culture microplate. The XF Cell Mito Stress Test was performed in glucose-containing XF base medium, following the manufacturer's protocol. Detailed protocols are available as Supplemental Information and at protocols.io (10.17504/protocols.io.v92e98e).

## Immunoblotting

Whole-cell lysates were prepared by incubating cells for 30 min in cold radioimmune precipitation assay buffer with Complete Protease Inhibitor Cocktail Tablets (Roche Diagnostics, Tokyo, Japan). Samples were then centrifuged at $10,000 \times$ g to allow the cell
debris to settle, and 30 μg total protein from the resulting supernatant was separated by 7.5% sodium dodecyl sulfate-polyacrylamide gel electrophoresis and electro-transferred onto membranes, probed with 1:500 (anti-Kv2.1 Ab) or 1:1000 (anti- β-actin Ab) of the indicated primary antibodies, and then probed with 1:8,000 of donkey anti-rabbit IgG (GE Healthcare, Piscataway, NJ) or sheep anti-mouse IgG (GE Healthcare) conjugated with horseradish peroxidase. Blots were visualized with enhanced Chemi-Lumi One Super (Nacalai Tesque, Kyoto, Japan) (*Daijo et al., 2016*; *Tanaka et al., 2011b*). Experiments were performed in triplicate, and representative blots are shown. Detailed protocols are available at protocols.io (10.17504/protocols.io.x9mfr46).

## Intracellular insulin concentration assay

Whole-cell lysates were prepared by incubating cells for 30 min in cold radioimmune precipitation assay buffer with Complete Protease Inhibitor Cocktail Tablets (Roche Diagnostics, Tokyo, Japan) after three time washing with PBS. Samples were then centrifuged at $10,000 \times g$ to allow the cell debris to settle to obtain total cell lysates. Then the concentration of insulin protein was assayed using the Mouse/Rat Insulin H-type enzyme-linked immunosorbent assay kit (Shibayagi Co. Ltd., Shibukawa, Japan). The concentrations were compensated with total protein weight.

## Quantitative reverse transcriptase-PCR analysis (*q*RT-PCR)

Total RNA was isolated using RNeasy Mini Kit (Qiagen, Valencia, CA, USA). First-strand cDNA synthesis and real-time PCR were performed as described previously (*Sumi et al., 2018a*). PCR primers are listed in Table S2. Detailed protocols are available in the Supplemental Information and from protocols.io (10.17504/protocols.io.x9mfr46).

## Electrophysiological studies

MIN6 cells were incubated in an extracellular bath solution containing 2 mM glucose at 37 °C for 30 min before patch-clamp experiments (*Hayashi et al., 2016*; *MacDonald et al., 2002*; *Zhang et al., 2016*). Membrane potential measurements and whole-cell current recordings were performed using the EPC 800 patch-clamp amplifier (HEKA Elektronik Inc. Holliston, MA, USA). Experiments were conducted at 23–30 °C. Detailed protocols are available in the Supplemental Information and from protocols.io (10.17504/protocols.io.v68e9hw).

## Statistical analysis

Data are presented as means $\pm$ standard deviation (SD). Differences between groups were evaluated by one-way analysis of variance (ANOVA) and two-way ANOVA followed by Dunnett's test or Tukey's test for multiple comparisons. Statistical analyses were performed with Prism7 (GraphPad Software, Inc. La Jolla, CA). Statistical significance was defined by $P$-values $<0.05$ (*Sumi et al., 2018a*).

## RESULTS

### Effects of propofol on insulin secretion

The response of insulin secretion to extracellular glucose concentration was examined in mouse MIN6 cells. The cells were cultured in 2 mM glucose and exposed to a range of glucose concentrations (2–20 mM) for 1 h. GSIS was detected in response to increased glucose concentrations (Fig. S1A). The time-dependent profile of GSIS was also investigated. The cells were exposed to 20 mM glucose and the GSIS was assayed at 0, 20, 40, 60, and 120 min (Fig. S1B). To compensate for the unknown influence of cell status on insulin measurement, insulin secretion is usually compensated with total protein weight. We measured the insulin concentration with or without compensation with the protein weight. As shown in Fig. S1C–S1E, the compensation did not impact insulin concentration measurement. Depending on the experimental data, we quantified insulin concentration without the compensation by total protein weight in this study in the case of MIN6 and INS1 cells. The insulin secretion in response to the polysaccharides sucrose and maltose was also assessed (Fig. S1F). Only glucose elicited insulin secretion, indicating other polysaccharides do not largely impact insulin secretion (IS).

Plasma concentrations of propofol in clinical settings during anesthesia and sedation are reported to range between 11 $\mu$M and 200 $\mu$M (*Ludbrook, Visco & Lam, 2002*; *Vanlander et al., 2015*). MIN6 cells were incubated with 5–100 $\mu$M of propofol in 2 mM glucose from 30 min to 4 h and then insulin concentrations were assessed. Although 5 $\mu$M and 100 $\mu$M propofol did not significantly increase basal IS, 10 $\mu$M propofol significantly increased IS after 1 h and 4 h incubation, 25 $\mu$M propofol significantly increased IS after 1 h incubation, and 50 $\mu$M propofol significantly increased IS after 30 min and 1 h incubation (Figs. 1A–1C). When cells were exposed to 20 mM glucose, 5 $\mu$M propofol did not significantly increase GSIS after any incubation period, whereas 10 $\mu$M propofol significantly increased GSIS after 1 h and 4 h incubation, 25 $\mu$M propofol significantly increased GSIS after 30 min and 1 h incubation and 50 $\mu$M propofol increased GSIS after 1 h incubation (Figs. 1D–1F). In contrast, 100 $\mu$M propofol significantly decreased GSIS compared to the control, after 1 h and 4 h incubation (Figs. 1D–1F).

Next, we tried to examine whether this GSIS induction by propofol was reversible. MIN6 cells were exposed to 25 $\mu$M propofol with 20 mM glucose for 1 h followed by incubation in 2 mM glucose without propofol for 6 h prior to re-exposure to 20 mM glucose under indicated concentrations of propofol (Fig. 1G). Statistically significant differences were not observed in GSIS of MIN6 cells with or without pre-treatment with 25 $\mu$M propofol. The evidence indicates that the enhancement effect of propofol on GSIS by less than 50 $\mu$M propofol is reversible. Next, we examined the effect of the isomer of propofol, 2,4-diisopropylphenol, which does not have hypnotic effects (*Tsuchiya et al., 2010*). We found that 5–50 $\mu$M 2,4-diisopropylphenol enhanced IS and GSIS, in a similar manner to propofol, after 1 h incubation (Figs. 1H and 1I). Therefore, our results demonstrate that the effect of propofol on GSIS is biphasic and is both dose- and time-dependent.

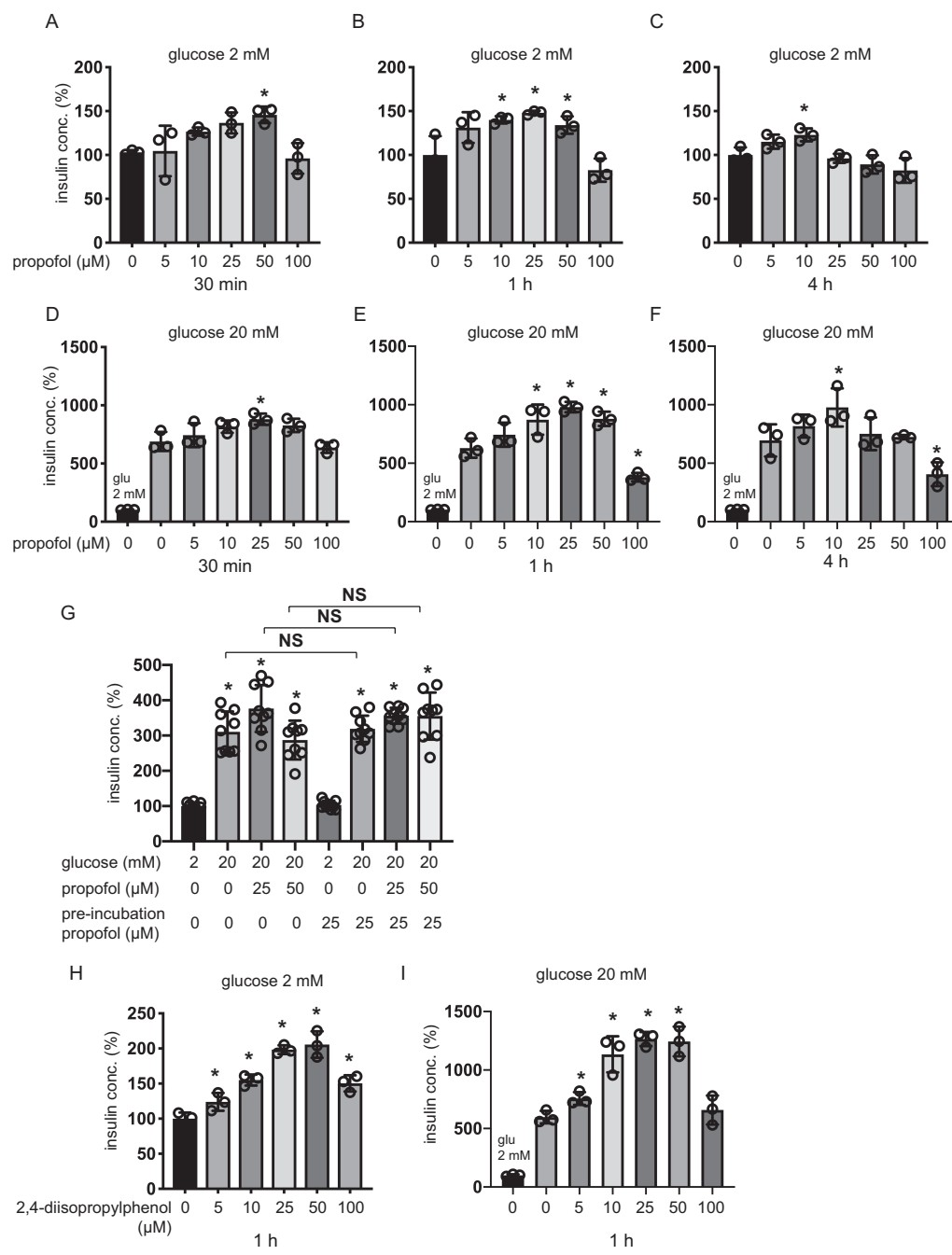

**Figure 1 Dose- and time-dependent effects of propofol on glucose-stimulated insulin secretion in MIN6 cells.** (A–C) Mouse MIN6 cells were exposed to propofol (0, 5, 10, 25, 50, and 100 µM) from 30 min to 4 h with 2 mM glucose. (D–F) Mouse MIN6 cells were exposed to propofol (0, 5, 10, 25, 50, and 100 µM) from 30 min to 4 h and then exposed to 20 mM glucose. (G) Mouse MIN6 cells were exposed to 0, 25, or 50 µM propofol with 2 mM glucose for 1 h; then cells were incubated with 2 mM glucose without propofol for 6 h prior to exposure to 20 mM glucose. (H, I) Mouse MIN6 cells were exposed to the propofol isomer 2,4-diisopropylphenol (5–100 µM) for 1 h in 2 mM and 20 mM glucose. Insulin secretion was determined as described in Materials and Methods. Data are presented as mean ± SD ($n = 4$). Differences between treatments were evaluated by one-way ANOVA followed by Dunnett's test for multiple comparisons. *$P < 0.05$, as compared with the control; #$P < 0.05$ for comparison of the indicated groups.

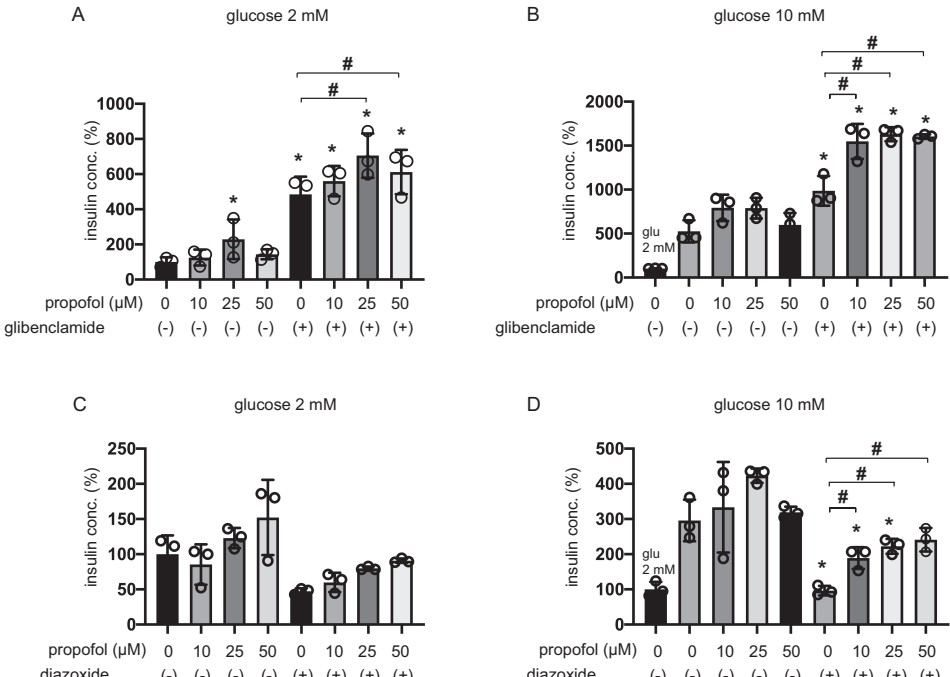

**Figure 2** Effect of propofol on insulin secretion induced by glibenclamide or inhibited by diazoxide.
(A, B) Mouse MIN6 cells were exposed to propofol (10, 25, and 50 μM) for 1 h with or without gliben-
clamide (100 μM) in 2 mM or 10 mM glucose. (C, D) Mouse MIN6 cells were exposed to propofol (10,
25, and 50 μM) for 1 h with or without diazoxide (100 μM) in 2 mM or 10 mM glucose. Insulin secretion
was determined as described in Materials and Methods. Data are presented as mean ± SD ($n = 4$). Dif-
ferences between treatments were evaluated by one-way ANOVA followed by Dunnett's test for multiple
comparisons. $^{*}P < 0.05$, as compared with the control (glucose = 2 mM, without glibenclamide or with-
out diazoxide treatment); $^{\#}P < 0.05$ for comparison of the groups indicated.

## Effects of propofol on insulin secretion induced by glibenclamide or inhibited by diazoxide

Elevation of intracellular ATP concentration ([ATPi]) in response to higher glucose
conditions closes ATP-sensitive potassium ($K_{ATP}$) channels and depolarizes the plasma
membrane (*Seino, 2012*). The channel closer glibenclamide facilitates insulin secretion
in pancreatic β-cells even in low-glucose conditions, while the channel opener diazoxide
inhibited insulin secretion in both 2 mM and 20 mM glucose (*Seino, 2012*).

The 25 μM and 50 μM propofol treatments significantly increased IS induced by 100 μM
glibenclamide in 2 mM glucose compared to control with no propofol or glibenclamide
treatment, whereas 10 μM propofol had no effect (Fig. 2A). In 10 mM glucose, propofol at
doses of 10 μM, 25 μM, and 50 μM enhanced GSIS induced by glibenclamide treatment
(Fig. 2B). On the other hand, 10, 25, or 50 μM propofol treatment alleviated 100 μM
diazoxide-elicited suppression of GSIS in 10 mM glucose (Figs. 2C and 2D).

The evidence strongly suggested that propofol impacts other pathways that are dependent
on $K_{ATP}$ channels sensitive to glibenclamide and diazoxide.

## Effects of propofol on GSIS in rat INS-1 cells and mouse pancreatic β-cells/islets

We next examined the effect of propofol on rat pancreatic β-cell-derived INS-1 cells. Propofol at doses of 25 μM enhanced IS under 2 mM glucose within 30 min, 1 h, and 4 h (Figs. 3A–3C). Under 20 mM glucose conditions, 25 μM propofol enhanced GSIS at all incubation periods (Figs. 3D–3F).

β-cells/islets were incubated with 25 μM of propofol in 2 mM glucose from 30 min to 1 h and then insulin concentration was assessed. In the case of β-cells/islets, insulin concentrations were compensated with total protein weight. As in the case of MIN6 and INS-1 cells, 25 μM propofol significantly increased IS after 1 h incubation (Fig. 3G). When cells were exposed to 20 mM glucose, 25 μM propofol significantly increased GSIS after 1 h incubation. Thus, propofol increased insulin secretion in only cell lines as well as primary mouse pancreatic β-cells/islets (Fig. 3G).

## Impact of propofol on proliferation and death of MIN6 cells

MIN6 cells growth under propofol treatment was examined using the MTS assay (Fig. 4A). Propofol treatment at 100 μM significantly suppressed the growth rate of MIN6 cells. The impact of propofol on cell death was also investigated. Camptothecin at 5 μM activated caspase 3/7 within 4 h, whereas 50 μM propofol did not activate caspase 3/7 within the same period. However, 100 μM propofol significantly activated caspase 3/7 at 4 h and 12 h incubation (Fig. 4B). Cell death was also assayed by the trypan blue exclusion method and was found to be significantly induced by 1 mM lidocaine. However, neither 50 μM nor 100 μM propofol induced cell death within the 12 h incubation period (Fig. 4C).

Our findings show that propofol at 50 μM did not affect proliferation or death of MIN6 cells, while propofol at 100 μM did after 12 h incubation.

## Effects of propofol on cellular energy metabolism

Intracellular ATP and ADP are generally assumed to play a crucial role in GSIS (*Rorsman, 1997*; *Seino, 2012*). Extracellular high glucose increases ATP in pancreatic β-cells (*Seino, 2012*). We investigated the effect of propofol on ATP under exposure to 2 mM or 20 mM glucose for 1, 4, and 8 h. With 2 mM glucose, 25 μM propofol did not affect ATP. In contrast, 100 μM propofol decreased ATP at 4 and 8 h exposure (Fig. 5A). With 20 mM glucose, 100 μM propofol decreased ATP at all exposure periods (Fig. 5B). Next, we investigated the effect of propofol on ADP (Fig. S2). In conditions where propofol did not affect ATP, the same concentration of propofol did not affect ADP as well.

The impact of propofol on oxygen metabolism of MIN6 cells was investigated. Stimulation with high glucose activated mitochondrial respiration in MIN6 cells. In 5 mM and 20 mM glucose, OCR increased in a time-dependent manner up to 4 h (Fig. S3A). ECAR also increased in a time-dependent manner (Fig. S3B). Neither 25 μM nor 50 μM propofol affected OCR in 20 mM glucose within the 4 h period of study. However, 100 μM propofol suppressed OCR (Fig. 5C). Therefore, our results show that clinically relevant doses of propofol do not affect cellular oxygen or energy metabolism.

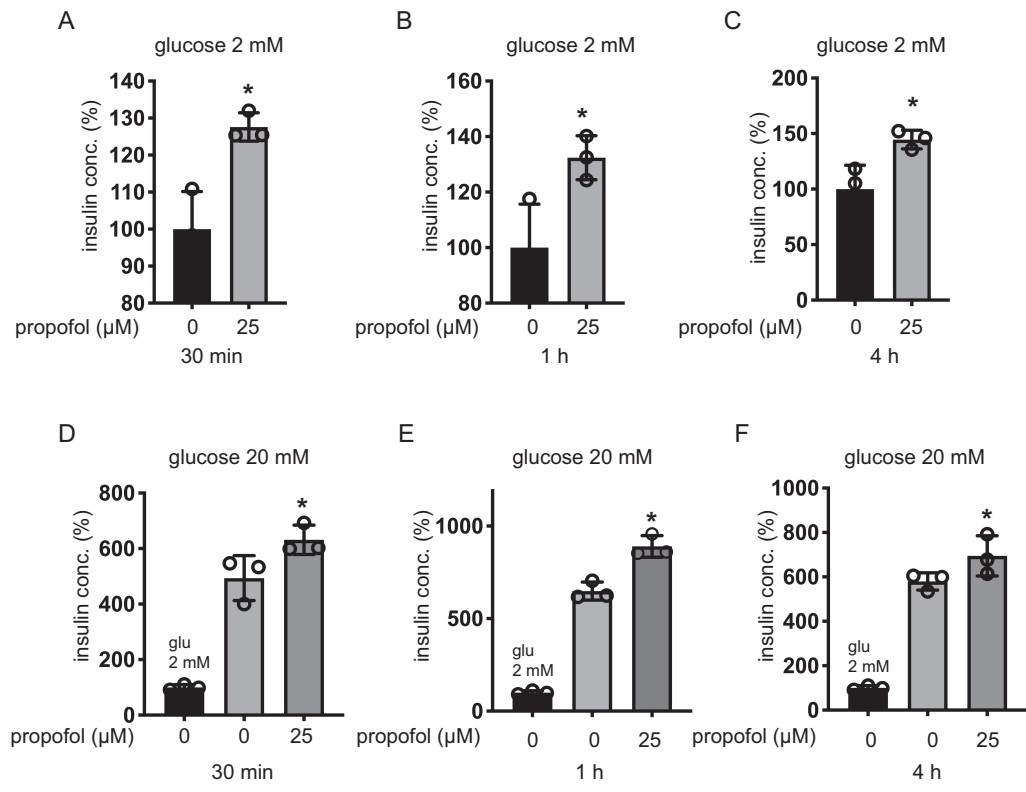

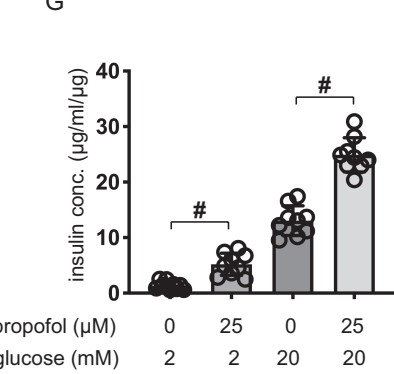

**Figure 3  Dose- and time-dependent effects of propofol on glucose-stimulated insulin secretion in INS-1 cells.** (A–C) INS-1 cells were exposed to 25 μM propofol from 30 min to 4 h in 2 mM glucose. (D–F) INS-1 cells were exposed to 25 μM propofol from 30 min to 4 h and then exposed to 20 mM glucose. (G) Mouse pancreatic β-cells/islets were exposed to 25 μM propofol for 1 h in 2 mM or 20 mM glucose. In the case of β-cells/islets insulin concentrations were compensated with total protein weight. Insulin secretion was determined as described in 'Materials and Methods'. Data are presented as mean ± SD ($n =$ 8). Differences between treatments were evaluated by one-way ANOVA followed by Dunnett's test for multiple comparisons. *$P < 0.05$, as compared with control (propofol 0 μM).

A

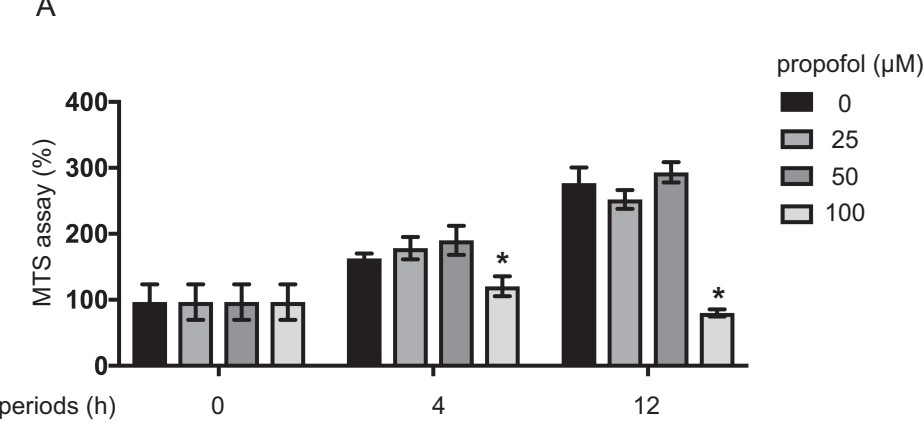

B

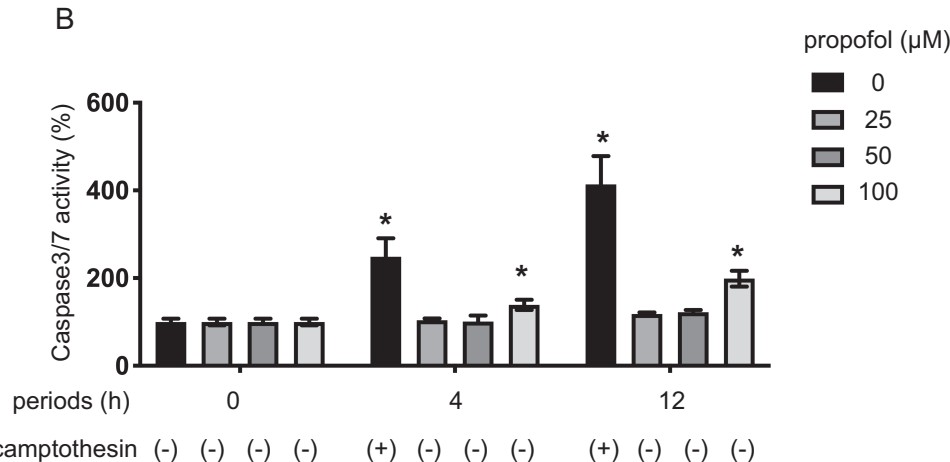

C

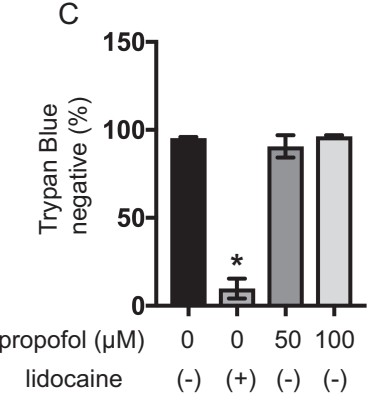

**Figure 4** **Impact of propofol on proliferation and death of mouse MIN6 cells.** (A, B) MIN6 cells were exposed to propofol at doses from 0 μM to 100 μM and to camptothecin at 5 μM and cultured for periods ranging from 0 h to 12 h prior to cell viability evaluation by MTS assay ($n = 3$) or caspase 3/7 activity ($n = 3$). Differences between treatments were evaluated by two-way ANOVA followed by Dunnett's test for multiple comparisons. *$P < 0.05$, as compared to the control cell population (continued on next page...)

**Figure 4 (…continued)**

[(A) propofol $= 0\,\mu$M, incubation time $= 0$ h; (B) propofol $= 0\,\mu$M, incubation time $= 0$ h, camptothecin $= 0\,\mu$M] (C) MIN6 cells were exposed to propofol at doses from $0\,\mu$M to $100\,\mu$M and lidocaine at 1 mM and cultured for 12 h. Differences between treatments were evaluated by one-way ANOVA followed by Dunnett's test for multiple comparisons. $^*P < 0.05$, as compared to the control cell population (propofol $= 0\,\mu$M, lidocaine $= 0$ mM treatment).

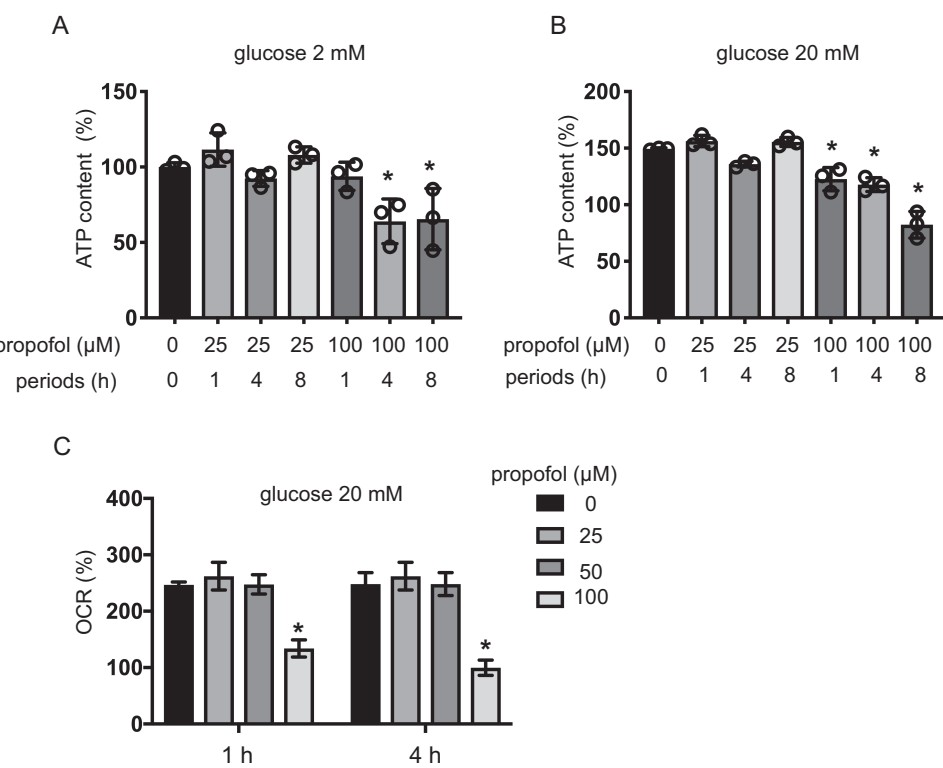

**Figure 5** **Effects of propofol on cellular energy metabolism.** (A, B) Mouse MIN6 cells were cultured from 1 h to 8 h with propofol at doses of $25\,\mu$M or $100\,\mu$M prior to determination of the cellular ATP level ($n = 3$) with 2 mM or 20 mM glucose. Differences between treatments were evaluated by one-way ANOVA followed by Dunnett's test for multiple comparisons. $^*P < 0.05$, as compared to the control cell population (propofol $= 0\,\mu$M, incubation period $= 0$ h treatment). (C) Mouse MIN6 cells were exposed to propofol at doses from $0\,\mu$M to $100\,\mu$M for a period of 1 h or 4 h, followed by oxygen consumption rate (OCR) assay. Differences between treatments were evaluated by two-way ANOVA followed by Dunnett's test for multiple comparisons. $^*P < 0.05$, as compared to the control cell population (propofol $= 0\,\mu$M).

## Impact of propofol and glucose concentration on expression of GLUT2, Cav1.2, Kir6.2, Kv2.1, SUR1, and insulin

The expression of glucose transporter 2 (GLUT2) channels including the voltage-dependent calcium channel Cav1.2, the Kir6.2 and SUR1 subunit of $K_{ATP}$, and the voltage-dependent potassium channel Kv2.1 and insulin was investigated after 1 h and 4 h incubation. The mRNA expression of these channels was not impacted by glucose concentration or by propofol within 1 h or 4 h (Figs. 6A–6F). Additionally, we investigate the mRNA expression

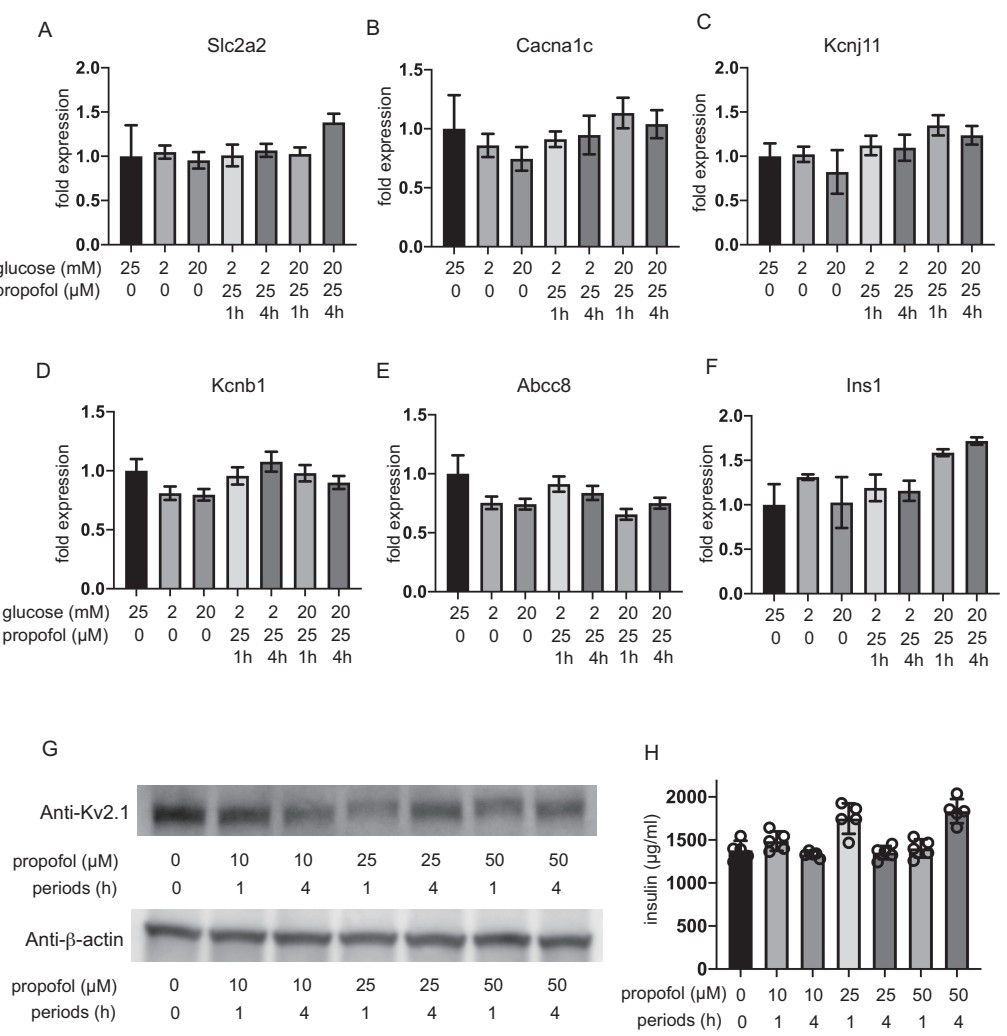

**Figure 6** **Impact of propofol and glucose concentration on the expression of glucose transporter 2, ion channels, and insulin.** Mouse MIN6 cells were exposed to 25 µM propofol with 20 mM glucose for 1 h and 4 h and harvested. Then the mRNA levels of Slc2a2(Glut2) (A), Cacna1c(CaV1.2) (B), Kcnj11(Kir6.2) (C), Kcnb1(Kv2.1) (D), Abcc8(SUR1) (E), and Ins1(Insulin) (F) were assayed by *q* RT-PCR. Data are presented as mean ± SD (*n* = 3). (G) The protein expression of Kv2.1 channel was investigated by immunoblot assay. (H) Intracellular insulin protein was investigated by ELISA.

at 12 h and 48 h. Incubation with 25 µM propofol did not affected the mRNA expression up to 48 h (Fig. S4).

Because the RT-PCR results clearly indicated that Kv2.2 mRNA was barely expressed in MIN6 cells compared to Kv2.1 (Fig. S5), we focused on evaluating the protein expression of Kv2.1. Immunoblot assay demonstrated the protein expression of Kv2.1 channel was not affected by propofol treatment (Fig. 6G and Fig. S6). Intracellular insulin contents were not affected by the treatment (Fig. 6H). The protein expression of Kv2.1 or insulin was not affected by 12 h or 48 h treatment as in the case of mRNA expression (Fig. S4).

These results demonstrate that the expression of molecules that play critical roles in glucose intake or membrane depolarization is not affected by propofol at clinically relevant doses.

## Effects of propofol on the membrane potential of MIN6 cells

We measured the membrane potential of MIN6 cells using gramicidin-perforated patch techniques to avoid dialyzing the intracellular components. In this configuration, the intracellular concentration of glucose is preserved, because the area of gramicidin-perforated patch membrane is much smaller than that of the whole-cell membrane. Propofol significantly increased the frequency of the action potential in 10 mM glucose from $0.32 \pm 0.15$ s$^{-1}$ to $0.84 \pm 0.42$ s$^{-1}$ at 10 $\mu$M and $1.08 \pm 0.43$ s$^{-1}$ at 25 $\mu$M (Fig. 7A; $n = 5$). Similar results were obtained with glibenclamide (100 $\mu$M). Application of additional propofol (25 $\mu$M) significantly increased the frequency of the action potential from $0.35 \pm 0.34$ s$^{-1}$ with glibenclamide to $0.59 \pm 0.23$ s$^{-1}$ (Fig. 7B; $n = 4$). Treatment with propofol also increased the duration of the action potential: $225.9 \pm 104.0$ ms with glibenclamide (i in Figs. 7B and 7C; $n = 4$) and $366.0 \pm 179.9$ ms with additional propofol (ii in Figs. 7B and 7D; $n = 4$). We measured whole-cell currents under the same conditions. The application of propofol (100 $\mu$M) decreased the slope conductance in a voltage range between $-83$ and $-63$ mV from $0.84 \pm 0.10$ to $0.23 \pm 0.12$ nS (Fig. 7E; $n = 3$). However, propofol had a negligible effect on the inward conductance. These results indicated that propofol affected the voltage-dependent outward conductance.

## Effects of propofol on voltage-dependent outward potassium currents in MIN6 cells

We measured whole-cell currents with K$^+$-rich pipette solution containing 3 mM ATP using voltage-clamp configuration and recorded voltage-dependent outward K$^+$ currents in MIN6 cells bathed in 10 mM glucose buffer (Fig. 8A). Typical traces in response to the Kv2.1/2.2-specific antagonist stromatoxin-1 (100 nM) are shown in Fig. 8B. Both peak and sustained K$^+$ currents were inhibited by stromatoxin-1 (Fig. S7, $n = 6$). Propofol at 50 $\mu$M significantly decreased K$^+$ currents from $137.0 \pm 19.4$ pA/pF to $91.1 \pm 22.4$ pA/pF at $+52$ mV (Fig. 8C, $n = 6$). The current–voltage relationships of sustained K$^+$ currents showed that propofol significantly inhibited the voltage-dependent outward K$^+$ currents in a dose-dependent manner (Fig. 8D). The typical traces in responses to propofol are shown in Fig. S8 . The Ki value for the effect of propofol was estimated at $83.6 \pm 13.0$ $\mu$M with a Hill coefficient of $1.3 \pm 0.3$ (Fig. 8E). The Ki value was not affected by voltage (Fig. 8F). Additionally, similar results were obtained with the isomer 2,4-diisopropylphenol. The Ki value for the effect of 2,4-diisopropylphenol was estimated at $41.1 \pm 5.4$ $\mu$M with a Hill coefficient of $1.5 \pm 0.2$ (Fig. S9).

We next examined the impact of stromatoxin-1 on IS of MIN6 cells (Figs. 8G–8H) and $\beta$-cells/islets (Fig. 8I) with both 2 mM and 20 mM glucose and Stromatoxin-1 affected IS in 2 mM glucose, and enhanced GSIS in 20 mM glucose conditions. Further, 100 $\mu$M stromatoxin-1 significantly increased IS even under 2 mM glucose conditions. Thus, in our experimental system using MIN6 cells and $\beta$-cells/islets, stromatoxin-1 increased IS as well as propofol.

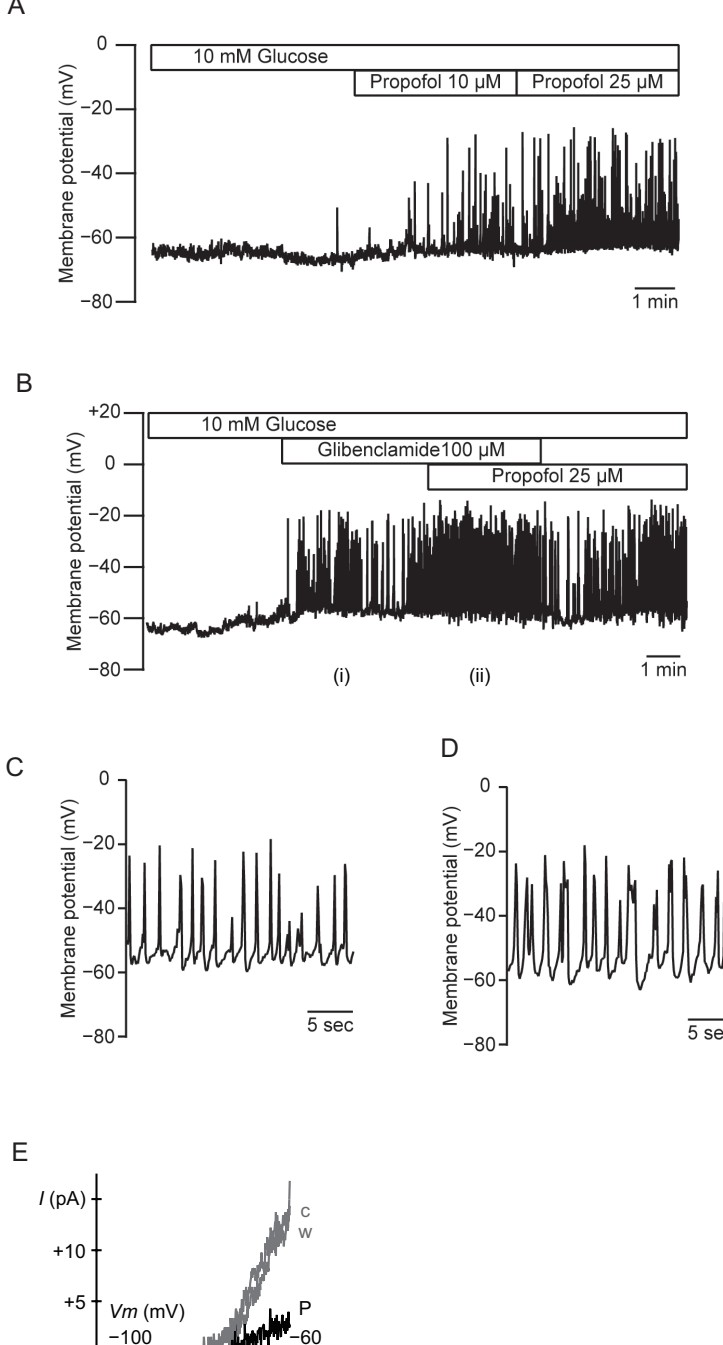

**Figure 7** **Effects of propofol on the membrane potential of mouse MIN6 cells.** (A) Propofol (10 and 25 μM) increased the frequency of the action potential with gramicidin-perforated patch (*n* = 5). (B) Gliben-clamide (100 μM) and additional propofol (25 μM) increased the frequency of the action potential (*n* = 4). The periods denoted as (i) and (ii) are shown in an expanded time scale as (C) and (D), respectively. (E) Representative current–voltage relationships for the whole-cell currents with gramicidin-perforated patch (*n* = 3). Propofol significantly decreased the voltage-dependent outward conductance. c, Control; w, wash-out; P, propofol 100 μM.

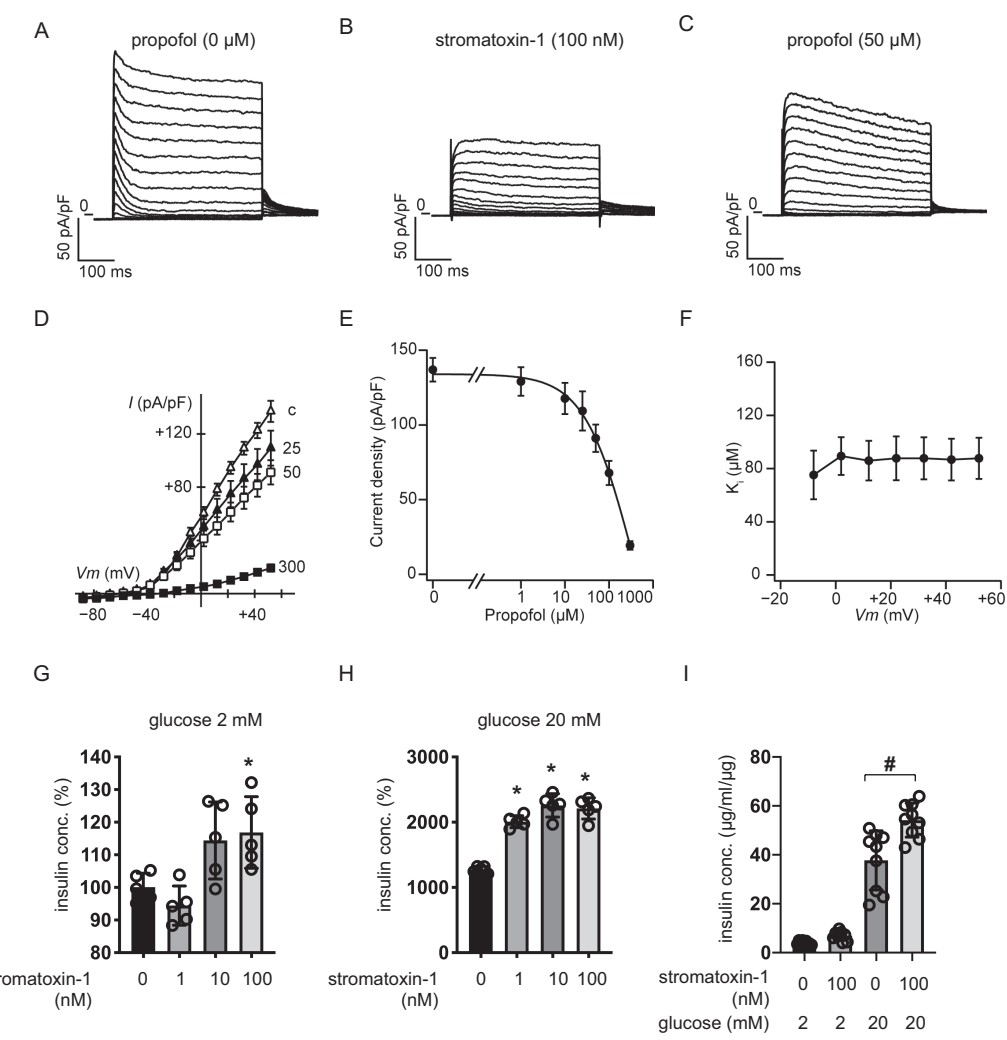

**Figure 8  Effect of propofol on voltage-dependent outward potassium currents in mouse MIN6 cells.**
Representative tracings of whole-cell currents obtained from a mouse MIN6 cell in the absence (A) or
the presence of stromatoxin-1 (100 nM) (B), and propofol (50 μM) (C). The cell was held at −88 mV
and stepped up for 0.4 s to potentials ranging between −88 and +52 mV in 10 mV steps. (D) Current–
voltage (I–V) relationships of whole-cell currents in the steady state with varying concentrations of propo-
fol (in μM). Propofol blocked the outward $K^+$ currents in a dose-dependent manner ($n = 6$). (E) Semi-
logarithmic plot of the current at +52 mV vs. concentration of propofol. The line is the best fit by the Hill
equation. (F) The inhibitory constant (Ki) was indifferent between −8 and +52 mV. (G, H) MIN6 cells
were exposed to strmatoxin-1 (0–100 nM) for 1 h with 2 mM and 20 mM glucose. (I) β-cells/islets were
exposed to stromatoxin-1 (100 nM) for 1 h with 2 mM and 20 mM glucose. Insulin secretion was deter-
mined as described in 'Materials and Methods'. Data are presented as mean ± SD ($n = 5$). Differences
between treatments were evaluated by one-way ANOVA followed by Dunnett's test for multiple compar-
isons. *$P < 0.05$, as compared with the control (stromatoxin-1 0 μM).

## DISCUSSION

In this study, we demonstrate that clinically relevant doses of the intravenous anesthetic propofol significantly enhanced insulin secretion under both basal IS and high glucose conditions (GSIS) in mouse and rat pancreatic β-cell-derived MIN6 and INS-1 cell lines and primary pancreatic β-cells/islets. We further found that this effect is dependent on inhibition of the stromatoxin-1-sensitive voltage-dependent potassium channel. Our estimate of the Ki value of propofol is 83.6 μM in a context of Kv channel inhibition. The concentrations of propofol we used in this study were from 5 to 100 μM. Plasma concentrations of propofol in clinical settings during anesthesia and sedation are reported to range between 2 μg/ mL (11 μM) and 5 μg/ mL (27.5 μM) (*Ludbrook, Visco & Lam, 2002*). Another study reported that the concentration of propofol in tissues of rats administered propofol at a dose of 20 mg/kg/h could reach 200 μM (*Vanlander et al., 2015*). The duration of exposure in our study ranged from 30 min to 4 h, which is also within clinically used periods of exposure (*Sumi et al., 2018a*; *Sumi et al., 2018b*). Therefore, both the propofol concentrations and the time range in this study are clinically relevant.

One of the intriguing findings of this study is the combined dose–time effect of propofol on IS and GSIS. Propofol at 25 or 50 μM enhanced IS at incubation periods of 30 min and 1 h. However, at 4 h incubation this amplifying effect of 25 and 50 μM propofol was not observed, with only 10 μM propofol enhancing IS. GSIS was also enhanced by 25 μM propofol at 30 min of incubation. On the other hand, longer incubation periods differentially affected GSIS. We previously reported that even clinically relevant doses of propofol suppressed the mitochondrial electron transfer chain and induced cell death by generating reactive oxygen species in SH-SY5Y cells (*Sumi et al., 2018b*). However, in MIN6 cells, 50 μM propofol did not induce caspase 3/7 activation or cell death within 4 h. Moreover, neither OCR nor ECAR was affected by 50 μM propofol. The evidence indicates that propofol at this concentration does not affect oxygen or energy metabolism. In contrast, 100 μM propofol suppressed oxygen metabolism and induced cell death, resulting in inhibition of IS. Therefore, propofol appears to affect IS in both a dose-dependent and a time-dependent manner.

Propofol is used intravenously as a hypnotic drug (*Sebel & Lowdon, 1989*). It exerts this effect through potentiation of the inhibitory neurotransmitter $\gamma$-aminobutyric acid (GABA) at the GABA$_A$ receptor (GABA$_A$R) (*Korol et al., 2018*; *Sebel & Lowdon, 1989*). β-cell-specific high-affinity GABA$_A$R subtypes and physiologically relevant GABA concentrations together modulate insulin secretion in human pancreatic islets (*Dong et al., 2006*; *Untereiner et al., 2019*; *Wang et al., 2019*). Several studies indicate the involvement of GABA in human islet-cell hormone homeostasis, as well as the maintenance of the β-cell mass. GABA exerts paracrine actions on $\alpha$ cells in suppressing glucagon secretion, and it has autocrine actions on human β cells that increase insulin secretion. GABA$_A$ receptor currents were enhanced by the benzodiazepine diazepam, the anesthetic propofol, and the incretin glucagon-like peptide-1 (GLP-1), but not affected by the hypnotic zolpidem in human islets β-cells (*Korol et al., 2018*). In contrast, one study indicated that exogenous GABA, baclofen (agonist of GABA$_B$ receptors), muscimol (agonist of GABA$_A$ receptors),

or bicuculline (antagonist of GABA$_A$ receptors) did not affect insulin release by isolated mouse or rat islets (*Gilon et al., 1991*). In this study, we used 2,4-diisopropylphenol, an isomer of propofol, which is not a GABA$_A$R ligand (*Tsuchiya et al., 2010*), and found that it also enhanced GSIS in a similar fashion as propofol. The evidence thus strongly suggested that GABA$_A$R is not involved in propofol-induced GSIS enhancement, at least in mouse or rat-derived β-cells.

A convincing model of GSIS has been established based on considerable experimental evidence (*Rorsman, 1997*; *Rorsman et al., 2000*; *Seino, 2012*). [ATPi] is generally assumed to play a crucial role in GSIS. The extracellular glucose concentration stimulates pancreatic β-cell metabolism and [ATPi] increases in β-cells. The activity of K$_{ATP}$ decreases in response to increased [ATPi]. The plasma membrane depolarizes to the threshold at which voltage-dependent calcium channels open. The influx of Ca$^{2+}$ facilitates exocytosis of insulin-containing vesicles.

Our results are in accordance with previous studies which show that treatment with propofol does not affect the expression of GLUT2, Kir6.2 or Cav1.2 (*Rorsman, 1997*; *Rorsman et al., 2000*; *Seino, 2012*). In this study, propofol at concentrations of less than 50 μM did not affect [ATPi], cell growth, or cell death. Oxygen metabolism was also not affected by propofol. Propofol inhibits K$_{ATP}$ channels overexpressed in COS-7 cells (*Kawano et al., 2004*; *Yamada et al., 2007*). However, our experimental results strongly suggest that glibenclamide- and diazoxide-sensitive K$_{ATP}$ channels are not involved in the modulation of IS and GSIS by propofol, at least in MIN6 cells.

Based on these results, we focused on voltage-dependent outward K$^+$ channels. Voltage-dependent outward K$^+$ currents in β-cells are reported to be involved in action potential repolarization, leading to limitation of Ca$^{2+}$ influx and insulin secretion. Indeed, previous studies show that the general Kv channel antagonist tetraethylammonium (TEA) augments membrane depolarization, Ca$^{2+}$ influx, and insulin secretion in a glucose-dependent manner (*MacDonald et al., 2002*; *Philipson et al., 1994*). Stromatoxin-1 inhibits Kv2.1 and Kv2.2, which encode delayed K$^+$ channels, with high affinities (*Chen, Kellett & Petkov, 2010*; *MacDonald et al., 2002*). Stromatoxin-1 is also a very sensitive inhibitor of Kv4.2, which encodes a transient K$^+$ current (*Escoubas et al., 2002*; *Wang & Schreurs, 2006*). In contrast, stromatoxin-1 has no effect on Kv1.1, Kv1.2, Kv1.3, Kv1.4, Kv1.5, Kv1.6, or Kv3.4 channels (*Escoubas et al., 2002*; *MacDonald et al., 2002*; *Wang & Schreurs, 2006*). Kv2.1, -4.1, -5.1, and -9.3 in addition to Kv11.1–11.3 were highly expressed in mouse MIN6 cells. In contrast, Kv2.2 and Kv4.2 were barely expressed in these cells (*MacDonald et al., 2002*). Three lines of evidence suggest that propofol selectively inhibits Kv2.1. First, MIN6 voltage-dependent outward K$^+$ currents were reduced by propofol in our study. Second, Kv2.1 protein is highly expressed and easily detectable in MIN6 cells and islets (*Hardy et al., 2009*). In contrast, Kv2.2 could barely be detected by RT-PCR at the mRNA level in MIN6 cells (*Hardy et al., 2009*). Third, propofol at 25 μM increased the frequency and duration of the action potential of MIN6 cells in the perforated patch-clamp configuration, suggesting that inhibition of the voltage-dependent K$^+$ conductance reduced the speed of repolarization (Fig. 7). Kv2.1 inhibition by stromatoxin-1 enhanced GSIS in MIN6 cells, as did 50 μM propofol. Importantly, glucose-stimulated membrane depolarization was

necessary to allow the insulinotropic action of specific Kv2.1 inhibition, since the effect is prevented by the $K_{ATP}$ channel agonist diazoxide (*MacDonald et al., 2002*; *Seino, 2012*).

One of the intriguing findings in this study is that propofol induced insulin secretion both under low and high glucose conditions. Our electrophysiological studies demonstrate that Kv2.1 is one of the target molecules for propofol. However, other studies indicate that Kv2.1 inhibition did not affect basal insulin secretion in mouse β-cells and MIN6 cells under low glucose conditions (*MacDonald et al., 2002*; *Zhou et al., 2018*). In our experimental system using MIN6 cells and β-cells, there was a tendency of stromatoxin-1 increasing IS in a dose-dependent fashion. The evidence may indicate that propofol impacts Kv2.1 channels to increase IS under 2 mM glucose conditions at least under our experimental systems. Otherwise, propofol may affect other Kv channels, $Ca^{2+}$-activated $K^+$ channels, and voltage-dependent $Ca^{2+}$ channels, as well as other cellular mechanisms such as exocytosis and metabolism. These ion fluxes set the resting membrane potential and the shape, rate, and pattern of firing of action potentials under different metabolic conditions. The $K_{ATP}$ channel-mediated $K^+$ efflux determines the resting membrane potential and keeps the excitability of the β-cell at low levels. $Ca^{2+}$ influx through Cav1.2 channels, a major type of β-cell Cav channels, causes the upstroke or depolarization phase of the action potential and regulates a wide range of β-cell functions, including the most elementary β-cell function, insulin secretion. $K^+$ efflux mediated by Kv2.1 delayed rectifier $K^+$ channels (a predominant form of β-cell Kv channels) brings about the downstroke or repolarization phase of the action potential, which acts as a brake for insulin secretion due to shutting down the Cav channel-mediated $Ca^{2+}$ entry. We demonstrated that the $K_{ATP}$ channel opener diazoxide suppressed basal insulin secretion under 2 mM glucose conditions. Moreover, 25 μM propofol increased basal IS under the same conditions and GSIS elicited by glibenclamide. The evidence suggests that $K_{ATP}$ channels are not a target of propofol.

Many studies clearly indicated that propofol inhibits human L-type calcium channels. Thus, it is not probable that propofol activates Cav1.2 channels (*Fassl et al., 2011*; *Olcese et al., 1994*; *Zhou et al., 1997*). Recently, several reports demonstrated that some transient receptor potential (TRP) channels are expressed in pancreatic β-cells and contribute to pancreatic β-cell functions (*Uchida & Tominaga, 2011*). It is reported that clinically relevant concentrations of propofol activated the recombinant transient receptor potential (TRP) receptors TRPA1 and TRPV1 heterologously expressed in HEK293t cells (*Fischer et al., 2010*). On the other hand, a report describes that propofol activates human and mouse TRPA1 but not human or mouse TRPV1 (*Nishimoto, Kashio & Tominaga, 2015*). Thus, involvement of TRP channels in promotion of basal IS by propofol seems if any to be restrictive. One study showed that, TASK-1, TASK-2, TASK-3, TREK-2, and TRESK-2 among two-pore domain $K^+$ ($K_{2P}$) channels were expressed in MIN6 cells (*Kang, Choe & Kim, 2004*). Another study revealed that pancreatic β-cell–specific ablation of TASK-1 channels augments glucose-stimulated calcium entry and IS (*Dadi, Vierra & Jacobson, 2014*). Previous reports demonstrate that TASK-1 and TREK-1 (TWIK-related acid-sensitive $K^+$ channel) are activated by volatile anesthetics (*Li et al., 2018*; *Putzke et al., 2007*). In contrast, propofol had no effect on human TASK-1 (or TASK-3) expressed

*Xenopus laevis* oocytes (*Putzke et al., 2007*). Although we did not test the possibility that propofol would affect $K_{2P}$ channels in cells derived from pancreatic β-cells, we cannot exclude this as a possible mechanism of propofol.

There is a discrepancy between the results we obtained in the insulin secretion studies and the results of the electrophysiological studies. In the insulin secretion studies, 100 µM propofol suppressed IS and GSIS. However, 100 µM and 300 µM propofol clearly inhibited Kv channels under whole-cell patch-clamp configuration. This discrepancy can be explained by the evidence that, at higher than clinically relevant doses, propofol combined with longer incubation times induces cell injury due to inhibition of the mitochondrial electron transport chain function (*Sumi et al., 2018a*; *Sumi et al., 2018b*). The impact of propofol on IS should thus be interpreted from the point of view of cell injury.

There are several limitations in this study. Although we identified stromatoxin-1-dependent Kv channels as a target of propofol in tumor-derived mouse MIN6, rat INS-1 and mouse pancreatic islets, studies on cells from human origin were not performed. Experiments using cells or islets from human origins can contribute to clinical application of these findings. We exclusively performed eletrophysiological studies using MIN6 cells. MIN6 cells are insulinoma cell line, which is derived from a transgenic mouse expressing the large T-antigen of SV40 in pancreatic β-cells. A line of studies indicates that the characteristics of MIN6 cells are very similar to those of isolated islets, indicating that this cell line is an appropriate model for studying the mechanism of glucose-stimulated insulin secretion in pancreatic β-cells (*Ishihara et al., 1993*; *Ishihara et al., 1995*; *Miyazaki et al., 1990*). Because we focused on the mechanism of propofol-induced insulin secretion in this study, we did not perform *in vivo* experiments. *In vivo* experiments may warrant the impact of propofol on systemic glucose metabolism. However, it is reported that glucose levels in rats anesthetized with sevoflurane were significantly higher than those in rats anesthetized with propofol, and insulin levels in rats anesthetized with sevoflurane were significantly lower than those in rats anesthetized with propofol (*Kitamura et al., 2012*). The evidence strongly suggests that anesthesia with propofol may suppress an increase in blood glucose by promoting insulin secretion.

## CONCLUSIONS

In conclusion, we have shown that propofol specifically blocks stromatoxin-1-sensitive voltage-dependent $K^+$ channels, probably inhibiting Kv2.1 currents and the inhibition results in insulin secretion in the presence of glucose in mouse MIN6 cells, rat INS-1 cells and mouse pancreatic β-cells/islets. Our data support the hypothesis that glucose induces membrane depolarization by closing $K_{ATP}$ channel, and blockade of Kv channels by propofol enhances depolarization, $Ca^{2+}$ entry, and insulin secretion in β-cells. Derivatives of propofol are potential candidates for the development of compounds that enhance and initiate β-cell electrical activity.

## ACKNOWLEDGEMENTS

We would like to thank Editage for English language editing.

### Funding

This work was supported by JSPS KAKENHI Grant Numbers JP26670693 and JP18K08877 to Kiichi Hirota and Grant Number JP16K10975 to Yoshiyuki Matsuo. It was also supported by a research grant B from Kansai Medical University to Chisato Sumi, a research grant from Kansai Medical University (KMU) research consortium to Kiichi Hirota, and a research grant from Katano Kai to Kiichi Hirota. The funders had no role in study design, data collection and analysis, decision to publish, or preparation of the manuscript.

### Grant Disclosures

The following grant information was disclosed by the authors:
JSPS KAKENHI: JP26670693, JP18K08877, JP16K10975.
Kansai Medical University.
Katano Kai.

### Competing Interests

The authors declare there are no competing interests.

### Author Contributions

- Munenori Kusunoki conceived and designed the experiments, performed the experiments, analyzed the data, prepared figures and/or tables, authored or reviewed drafts of the paper, approved the final draft.
- Mikio Hayashi conceived and designed the experiments, analyzed the data, prepared figures and/or tables, authored or reviewed drafts of the paper, approved the final draft.
- Tomohiro Shoji, Takeo Uba, Hiromasa Tanaka, Chisato Sumi performed the experiments, authored or reviewed drafts of the paper, approved the final draft.
- Yoshiyuki Matsuo conceived and designed the experiments, analyzed the data, contributed reagents/materials/analysis tools, authored or reviewed drafts of the paper, approved the final draft.
- Kiichi Hirota conceived and designed the experiments, analyzed the data, authored or reviewed drafts of the paper, approved the final draft.

### Data Availability

The raw data is available in the Supplemental Files.

### Supplemental Information

Supplemental information for this article can be found online at http://dx.doi.org/10.7717/peerj.8157#supplemental-information.

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
