# Peer review of "Propofol inhibits stromatoxin-1-sensitive voltage-dependent K+ channels in pancreatic β-cells and enhances insulin secretion"

_PeerJ, doi:10.7717/peerj.8157_

## Round 0.1 · original submission · Major Revisions

Although the paper is interesting, there are several points to be addressed, as indicated by the reviewers. First, the authors should prove their studies in murine isolated islets/beta cells and/or in vivo, in mice, as continuous cell lines have important limitations. Literature citation should be expanded. English should be thoroughly improved.

Reviewer 1 ·

Basic reporting

This paper investigated the mechanism by which propofol induce insulin secretion. Because propofol is used in perioperative patients and that glucose control is important in such patients, the data shown in this study is potentially interesting and beneficial. However, the paper is missing several critical information.

Experimental design

The most critical point of this study is that (as authors also mentioned in discussion as limitation), authors only performed experiments in cell line. This reviewer does not understand why authors used both MIN-6 and INS-1. Authors should simply isolate islets from mice and perform batch incubation for insulin secretion and use primary beta cells for electrophysiological study.

Also, measurement of blood glucose in vivo under propofol treatment would be critical information that should be included in this paper. If propofol affects short term insulin secretion, it should show immediate change in glucose level in mice.

Also it is unnecessary to perform insulin secretion experiment in cell line using sucrose and maltose. Did authors wanted to compensate osmotic pressure of incubation solution? If so, it should be noted in the manuscript.

This reviewer do not understand why authors measured SUR2 for qRT-PCR assay (which did authors measured SUR2A or 2B? or both???). Authors should know that beta-cell KATP channel is composed of Kir6.2 and SUR1, not SUR2.

Because propofol induced insulin secretion in low glucose condition, the underlying mechanism for insulin secretion by propofol is unlikely to be only by the Kv channel current inhibition. In low glucose concentration (in hyperpolarized state), Kv channel is closed so propofol cannot enhance insulin secretion through inhibition of Kv channel. Also, there is confusion in description about contribution of KATP channel in this paper. In Result section (line 241), authors mentioned that "This evidence suggests that propofol affects ion channels such as ATP-sensitive potassium channels which are otherwise insensitive to glibenclamide and diazoxide (???, very confusing description and hard to understand....)" but in discussion section, authors mention"KATP channels are not involved in modulation of IS and GSIS by propofol (line 373)". This is very confusing. In order to clarify this, authors should simply measure the KATP channel current with and without propofol in both whole cell and perforated patch mode.

Validity of the findings

If authors can show more clearly about underlying mechanism for insulin secretion induced by propofol, this study is very interesting and clinically important.

Additional comments

The study itself is interesting and contain important information. However, in the present form, this study can only say that propofol can affect Kv channel activity in cell line. It is not clear whether this Kv channel modulation truly affects insulin secretion. With present data there is still a possibility that propofol is affecting KATP channel or any other factor that may influence insulin secretion.

Reviewer 2 ·

Basic reporting

In this manuscript the Authors studied the effect of Propofol, an intravenous anesthetic drug, on insulin secretion and suggests that it may act modulating potassium channel activity. However, there are some aspects that need to be clarified and reviewed.
Minor point:

- In the title, the Authors mention MIN6 cells as the in vitro model of pancreatic beta cells under study. However, INS-1 cells were studied as well., A more generic title would be appropriate to introduce the reader to the manuscript

Experimental design

Many experimental aspects need to be improved. My comments are as follows:
- In my opinion it is not necessary to use different doses of propofol in each experiment. After doing an initial dose response evaluation, the more effective dose should be used. In the specific case, I believe that 25 µM may be fine. It is also somehow concerning that the same doses give different results in different experiments (compare for example Fig. 1a; 2a; 2c)
- Why did the Authors choose 100 µM propofol to study cellular energy metabolism (fig. 5) if they previously demonstrated a negative effect of this dose on insulin secretion? and why did the Authors use two other different doses of Propofol to evaluate oxygen metabolism (fig 5)? As I suggested before, one Propofol dose should be chosen and used for all the evaluations.
- Fig. 6., 1h of Propofol treatment is not sufficient to evaluate gene expression variations. Please, consider a longer time to investigate the effect of Propofol on gene expression. The best choice would be a time course assay.
- Fig 6., why have the authors performed western blot experiments only for Kv2.1 ? Protein expression levels of the other targets may be also worth investigating. However, a longer treatment time would be preferable to investigate changes in protein expression levels.
- Lines 296, 297, 298,299, 300. “Propofol increase the frequency of action potential in 10mM glucose from 0.32 to 0.84 at 10 µM and to 1.08 at 25 µM. Similar results were obtained with glibenclamide (100 µM). Application of additional propofono (25 µM) increased the frequency of the action potential from 0.35 with glibenclamide to 0.59 (fig. 7)…” It does not seem that adding propofol improves the increment obtained when glibenclamide was used alone. Please, clarify this aspect.

Validity of the findings

The results obtained are really interesting and could have important clinical implications, since propofol and other anesthetic drugs are widely and routinely used in clinical practice every day. Indeed, if these data are confirmed, physicians should consider the hypoglycemic risks to which patients would be exposed. In this regard, I suggest the Authors to emphasize the importance of this aspect in the text.

Reviewer 3 ·

Basic reporting

The authors investigated propofol effects on glucose-induced insulin secretion, and they try to dissect the mechanism of action of this anaesthetic drug. The claim that propofol specifically blocks specifically stromatoxin-1-sensitive voltage-dependent K+ channels, by blocking Kv2.1 currents in MIN6 cells, and that the inhibition increases insulin secretion in response to glucose. They also do some test on INS-1 cells.
Literature citing is limited. It would be desirable to review other authors too.
The English language is not always clear, should be reviewed.
Expression of results is not convincing if the authors do not make clear how were the percentages calculated. It would be more convincing if they give the raw data in the bars.
One of the main problems of this manuscript is that they take MIN-6 cells as normal beta cells, and this is not exact. One of the main differences is that cell lines divide, while healthy cells do not. This concept is taken very far because, in the discussion, they consider that the effects they observed could be significant for the anaesthetic procedures in humans! They did not even took a mouse to check the facts in the whole animal. The model should develop hypoglycemia.

Experimental design

The experimental design was easy because they use cell lines, but it would be better to check the results in the whole animal, and single real beta cells.

In the experiments in the whole cell mode of the patch clamp technique is essential to take in consideration that glucose cannot be metabolised, cells are dialysed. Glucose will not affect currents under these conditions.

MIN6 cells do not respond too much to glucose; they could be in different moments of the cell cycle. The effect of propofol is variable. The drug without hypnotic effect also increases IS and GSIS, we could assume a not specific observation.

I would not expect too many changes in GLUT2 or the other channels expressed in the membrane of the cells after one-hour exposition to propofol.

Validity of the findings

In the experiments with glibenclamide and propofol, you conclude that propofol affects channels that are insensitive to glibenclamide; while there could be other explanations, it could be affecting calcium currents or other IS mechanisms.

They must discuss the limitations of the study.

Additional comments

The results are too many and expressed as a percentage of something that is not clear, are difficult to follow.
When you comment on the results in the text it would be more useful to manage as the percentage increase or decrease, better than the numbers that are already in the graphics.
The discussion could be more profound.

---

## Round 0.2 · Major Revisions

Issues raised by reviewers 2 and 3 were not fully addressed.

English editing of the text should have included the newly added parts: this should be seriously addressed as well.

Reviewer 1 ·

Basic reporting

Authors have improved according to my suggestion.

Experimental design

Designs are accurate and I have no further comment.

Validity of the findings

Interesting enough to be published.

Additional comments

I have no further comments.

Reviewer 2 ·

Basic reporting

...

Experimental design

...

Validity of the findings

...

Additional comments

The Authors have not responded satisfactorily to my previously comments. In particular, my observations are as follows:

1) The Authors write that the choice of 100uM of propofol was made because this concentration had a negative effect on insulin secretion both under 2mM glucose and for GSIS. However, the aim of this work was to evaluate the impact of propofol on insulin secretion at low and high glucose levels. For this reason, it would be preferable to use 25uM propofol to evaluate its effects on cellular energy metabolism.
2) In my opinion, a time course with a range between 1 hour and 4 hour it is no enough to assess effects on expression levels. Please, set a time course at least up to 12 hours for the gene expression study and at least up to 48 hours to study protein expression.
3) In my previous comment I did not mention Kv2.2 channel subunit. In fig. 6., the Authors show the impact of propofol and glucose concentration on the expression of glucose transporter 2, ion channels and insulin. Western blot was performed only for Kv2.1 protein. Why only for Kv2.1? Protein expression levels of Slc2a2, Cacna1c, Kcnj11, Kcnb1, Abcc8 and Ins1 may be also worth investigating. However, as stated in point 2, a longer and appropriate treatment time (at least 48hours) would be preferable to investigate changes of protein levels.

Reviewer 3 ·

Basic reporting

This is a second review. I only want to state that the title must state tumoral cell lines because most of the work was done on them, and they have many differences from mice cells.

Experimental design

No more comments

Validity of the findings

Please comment on all the issues of the first review.

Additional comments

The manuscript is better, but I still have some doubts.
I want to see a mean IV curve, with the number of cells studied. They are still MIN cells instead of normal mouse cells. I do not understand this procedure.
The effects on the basal insulin secretion are severe, and that could cause hypoglycemia. I suggested this possibility in my first review, but you did not pay attention to it. The discussion is still missing that they work mostly on tumoral cell lines; you must give the limits to the study.
There are still many orthographic mistakes, especially on the corrected part of the manuscript.

---

## Round 0.3 · Minor Revisions

The authors have satisfactorily addressed the Reviewers' concerns. However, as indicated by Reviewer 2, please modify Fig. 6 accordingly.

Reviewer 2 ·

Basic reporting

...

Experimental design

...

Validity of the findings

...

Additional comments

New figure 6 is not clear. I think that there has been a misunderstanding. The Authors have changed the terms of comparison of the gene expression. Indeed, in the previous figure all data were compared to 2 mM glucose / 0 uM propofol. In the new version of the figure the Authors compare all points to 25 mM glucose / 0 uM propofol. Furthermore, since the Authors want also to investigate the effect of glucose on the expression genes, I suggest them to prepare two different graphs: in one they can show the effect of glucose on gene expression and in the other one the combined effect of glucose and propofol until 48h. Please, in this second graphs leave also original data with 1 and 4 hours expression levels (figure 6 of the previous version). Judging from the histograms height, it seems that there is some change during the time course.

---

## Round 0.4 · accepted · Accept

The authors have satisfactorily fulfilled all the reviewers' requests.